# High Adherence to a Mediterranean Alcohol-Drinking Pattern and Mediterranean Diet Can Mitigate the Harmful Effect of Alcohol on Mortality Risk

**DOI:** 10.3390/nu16010059

**Published:** 2023-12-24

**Authors:** Angelo Campanella, Caterina Bonfiglio, Francesco Cuccaro, Rossella Donghia, Rossella Tatoli, Gianluigi Giannelli

**Affiliations:** 1National Institute of Gastroenterology—IRCCS “Saverio de Bellis”, 70013 Castellana Grotte, Italy; catia.bonfiglio@irccsdebellis.it (C.B.); rossella.donghia@irccsdebellis.it (R.D.); rossella.tatoli@irccsdebellis.it (R.T.); gianluigi.giannelli@irccsdebellis.it (G.G.); 2Local Health Unit—Barletta-Andria-Trani, 76121 Barletta, Italy; francescocuccaroepi@gmail.com

**Keywords:** alcohol, Mediterranean Diet, mortality risk

## Abstract

Background: Alcohol is a psychoactive substance with deleterious effects on human health and mortality. This study aims to investigate the joint associations between the Mediterranean Diet (MedDiet), alcohol- consumption patterns and mortality from the following: all causes, cardiovascular, neoplastic, the digestive system, and other causes. Methods: A sample of 3411 alcohol consumers aged ≥18 years was selected from two prospective cohort studies: the MICOL and NUTRIHEP Study. Cohorts were enrolled in 2005–2006, and followed up until December 2022, capturing data on alcohol consumption, diet, and mortality. Adherence to the MedDiet was measured by the relative Mediterranean score (rMED), and alcohol consumption by the Mediterranean Alcohol-drinking Pattern index (MADP). Statistical analyses included flexible parametric survival models and subdistribution hazard ratios, to consider different causes of death. Results: a significant increase in digestive-system (SHR 2.77, 95% CI 1.16; 63) and cancer mortality risk (SHR 2.25, 95% CI 1.08; 4.70) was observed among individuals with low adherence to the MADP. Low adherence to the Mediterranean pattern of alcohol consumption, combined with low adherence to the MedDiet, was associated with higher overall mortality (HR 2.29, 95% CI 1.04, 5.04), and, in particular, with higher mortality from digestive system diseases (SHR 4.38, 95% CI 1.22, 15.8). Conclusions: This study suggests that deleterious effects of alcohol on mortality vary, depending on alcohol consumption patterns and dietary context. Higher adherence to the MedDiet appears to mitigate the adverse effects of moderate alcohol consumption, particularly for wine drinkers.

## 1. Introduction

Alcohol, as an ethanol, is a psychoactive substance with a significant impact on public health and mortality. Several epidemiological studies have described the J-shaped or U-shaped relationship between alcohol intake and general and cardiovascular mortality, suggesting a beneficial effect of low-to-moderate consumption [1,2,3].

Alcohol has also been described as an important risk factor for the development and mortality of several types of cancer, in particular lip and oral-cavity cancer, nasopharynx cancer, other pharyngeal cancers, esophageal cancer, larynx cancer, colon and rectum cancer, breast cancer, and liver cancer [4,5].

Chronic alcohol abuse and overconsumption have been associated with many health problems, such as cardiovascular events [6] and liver disease [7]. Moreover, alcohol is often involved in traffic accidents, domestic violence and other dangerous behaviors that can lead to premature death [8].

The effect of alcohol in moderate quantities, however, remains controversial. A safe and acceptable level of alcohol intake would not appear to exist, and the effects recorded in the past as cardioprotective with minimal or moderate alcohol use are confounding, because they are burdened by lifestyle and socio-demographic factors [9].

Consumption patterns and the context in which alcohol is drunk may therefore influence the effects of alcohol. Recent studies have shown that the population-level health risks associated with low levels of alcohol consumption vary from region to region, and are greater for younger than for older populations [10].

Moderate amounts of alcohol are characteristic of the Mediterranean dietary pattern, whose effect on longevity has been extensively studied [11]. Alcohol in the Mediterranean countries is mostly consumed during meals, and mainly in the form of wine. Recent cohort studies carried out in Spain have shown that higher adherence to the Mediterranean alcohol-drinking pattern (MADP) index may be protective against general mortality outcomes [11,12]. 

The MADP is an index that captures various aspects of alcohol consumption relative to typical Mediterranean alcohol consumption behaviors: quantity, type of beverage (e.g., spirits, wine), and pattern of consumption [13].

Our objective was to investigate the effect of alcohol consumption patterns assessed with the MADP index and Mediterranean diet (MedDiet), and their joint interaction on all-cause mortality, which includes cardiovascular disease, digestive disorders, cancer and other causes. The observed study population is part of a large prospective cohort from Southern Italy, and the vital status was followed up between January 2006 and December 2022.

## 2. Materials and Methods

This work included data from two different prospective cohort studies conducted by the Laboratory of Epidemiology and Biostatistics of the National Institute of Gastroenterology, Research Hospital ‘Saverio de Bellis’ (Castellana Grotte, Bari, Italy). Details about the study population have been published elsewhere [14,15].

In summary, the Multicentrica Italiana Colelitiasi (MICOL) Study is a population-based prospective cohort study of subjects randomly drawn from the electoral list of Castellana Grotte (≥30 years old) in 1985, and followed up in 1992, 2005–2006 and 2013–2016. In 2005–2006, this cohort was integrated with a random sample of participants from the PANEL study, aged 30–50 years, to compensate for cohort aging.

The Nutrition Hepatology (NUTRIHEP) Study is a cohort of subjects from the city of Putignano (Apulia, Southern Italy), extracted from the 2005–2006 medical records of general practitioners in Putignano (≥18 years). We used general practitioners’ (GPs’) medical records instead of drawing from the census, because no significant difference was observed between the distribution of the general population from Putignano and the subjects in the GPs’ records. 

Considering the period 2005–2006 as the baseline for both studies, in total, 5271 subjects were invited to participate, among which 3411 alcohol consumers were included in the analysis (Figure 1).

All procedures were conducted in accordance with the ethical standards of the institutional research committee (IRCCS Saverio de Bellis Research and the Ethical Committee approval for the MICOL Study (DDG-CE-347/1984; DDG-CE-453/1991; DDG-CE-589/2004; DDG-CE 782/2013); the NUTRIHEP Study in 2005 and 2014 (DDG-CE-502/2005; DDG-CE-792/2014), and with the 1964 Helsinki declaration and later amendments.

### 2.1. Data Collection

Participants in the MICOL and NUTRIHEP studies underwent anthropometric measurements, blood sampling and ultrasonography of the liver. They were weighed on an electronic scale, SECA, wearing underwear, and weight was approximated to the nearest 0.1 kg. Height was measured with a SECA wall stadiometer, approximated to the nearest 1 cm. Blood pressure measurements were performed according to international guidelines [16].

The mean of 3 blood pressure measurements was calculated. Fasting venous blood samples were drawn, and the serum was separated into two different aliquots. One aliquot was immediately stored at −80 °C, and the second aliquot was used to test biochemical serum markers using standard laboratory techniques, in our Central Laboratory.

Participants were interviewed by trained operators to collect information on sociodemographic characteristics, health status, personal history and lifestyle factors including history of tobacco use (never, former—quit 5 or more years before—and current), education (none, primary school, secondary school, high school, graduate) [17] job (managers and professionals, crafts, agricultural and sales workers, elementary occupations, housewives, pensioners and unemployed) [18] and marital status (single, married/cohabiting, separated/divorced and widow/er).

Trained nutritionists administered the European Prospective Investigation on Cancer Food Frequency Questionnaire (EPIC), to estimate the usual food and alcohol intake. Individual nutrient intakes were derived from foods included in the dietary questionnaires through the standardized EPIC Nutrient database. The EPIC questionnaire input was performed online, and centralized processing was carried out by the National Cancer Institute, based in Milan [19].

### 2.2. Outcome Assessment

We investigated the combined effect of alcohol consumption and adherence to the MedDiet on causes of death, including deaths from cardiovascular disease, cancer and other causes. Participants were followed up until 31 December 2022, and the vital status or any emigrations were verified through the registry office of the municipalities of Castellana Grotte and Putignano. 

Information on causes of death from 2006 to December 2022 was extracted from the Apulian Regional Registry, using the death certificate, according to WHO guidelines [20].

Causes of death were grouped as follows: cardiovascular disease (CVD) (ICD-10 I00-I99), overall cancer without gastrointestinal cancer (Cr) (ICD-10 C30-C97), digestive system (DS), which included codes K25.9; K55.9; K56.4; K.56.6; K57.2; K63.8; K72.9; K73.9; K74.6; K76.9; K81.0; K86.1; K86.8; and K92.2, viral hepatitis (B16; B17; B18) and gastrointestinal cancer (C00–C26). Deaths from other causes (DOC) included the remaining ICD-10 codes. 

### 2.3. Exposure Assessment

Adherence to the MedDiet was evaluated with the Mediterranean relative scoring system [21], which assigns scores (rMED) without alcohol intake (rMEDNA) based on the tertiles of reported intake of nine distinct food clusters, including fruit (excluding fruit juices), vegetables (excluding potatoes), fresh fish, cereals, legumes, olive oil, meat and dairy products.

A score of 0, 1 or 2 (with 2 as the maximum score) was assigned for increasing tertiles of each food cluster, with the exception of meat and dairy products, which were assigned a score of 2, 1 or 0 as the maximum score. The scores were summed, yielding rMEDNA scores ranging from 0 to 18, and classified as low (0–6), medium (7–10) and high (11–18) adherence, as suggested by the literature [21].

The advantage of this scoring system is that it is ‘energy-adjusted’ (based on grams of a food group per 1000 kcal/day), classified on the basis of tertiles of intake of each component (except alcohol), and includes the intake of olive oil, of which Italy is a major producer and consumer [22].

Consumption of alcoholic beverages and other information related to the consumption of alcoholic beverages during the years preceding enrolment was collected, using the EPIC questionnaire.

In the main analysis we excluded the group of abstainers, because we could not distinguish between lifelong abstainers (by personal choice) and those who stopped consuming alcohol for health reasons. In the Appendix A we show the results that included abstainers in the analysis. (Appendix A)

Alcohol consumption pattern was assessed with adherence to the MADP. The scoring system was built up referring to the MADP indications applied to the SUN cohort study [13]. The score consisted of 7 items linked to the following domains: (1) moderate alcohol intake (g/d), (2) alcohol consumption per week (d/week: week ratio), (3) low spirit consumption (alcohol from spirits/total alcohol), (4) wine preference (alcohol from wine/total alcohol), (5) wine consumed preferably with meals (wine with meals/total wine), (6) preference for red wine over other types of wine (red wine/total wine), and (7) no excess consumption (maximum drinks on a single occasion).

The first two items scored 0 to 2, while the remaining five scored 0 to 1. Higher scores indicate greater adherence to the MADP [13]. Information about physical activity was unavailable.

### 2.4. Statistical Analysis

For analytical purposes, the MADP was grouped into three categories: low adherence (0–3 score), moderate adherence (4–6 score), and high adherence (7–9 score). 

Gea. et al. [13] had divided the category moderate adherence (4–6) into moderate–low (3–4) and moderate–high (5–6). However, in order to make the representation of the sample more homogeneous, we grouped the subjects with moderate adherence into a single category (moderate adherence, 4–6 score). Cut-off points to assign the score to each of the seven items assessed were taken from other studies [11,23].

We considered nine categories obtained by combining MADP and rMEDNA for analysis. Data are presented as mean (±SD), median (±IQR) for continuous data, or frequency (%) for categorical data. ANOVA and Pearson’s chi-square tests were used to test differences between means and proportions.

The observation time was from enrolment to death, moving elsewhere, or end of the study (31 December 2022), whichever occurred first. Since age is the most important risk factor for death, we chose age at death as the time scale.

We set 90 years of age as the maximum observation age, to avoid problems related to comorbidities in the over-90s and worse coding quality of deaths occurring in very old age. A flexible parametric model was fitted to the data, to assess the association between MADP, rMEDNA and MADP + rMEDNA and all-cause mortality. In this work, we used high adherence as the reference group. Schoenfeld residuals were performed to test proportional hazards assumptions.

For cause-specific mortality, proportional hazard models were run for sub-distributions, using a competing risk approach [24]. We estimated the sub-distribution hazard ratio (SHR) for the MADP, rMEDNA and MADP + rMEDNA association with the risk of developing four types of competing events: CVD, Cr, DS, and death from other causes. Finally, using post-estimation tools, we predicted SHRs by cause [25]. 

Confounding variables were selected, based on the existing literature (Charlson Comorbidity Index (CCI), BMI, smoking habits, and gender) [11] or based on the lowest value of Akaike’s criterion (AIC) and Schwarz’s criterion of Bayesian information (BIC) (Glutamate Pyruvate Transaminase (GPT), Triglycerides (TGL), and job) [26,27].

All statistical analyses were performed using Stata, statistical software version 18.0 (StataCorp, 4905 Lakeway Drive, College Station, TX 77845, USA); in particular, the stpmcr2 official Stata command and its post-estimation commands were used.

## 3. Results

A total of 592 (17.4%) subjects died during the observation time (227,370.48 person-years), 175 (29.5%) from cardiovascular diseases, 111 (18.75%) from neoplastic diseases, 83 (14%) from digestive system diseases and the remaining 223 (37.6%) from other causes.

The main characteristics of the 3411 participants, classified according to adherence to MADP, are shown in Table 1; 1810 subjects had high adherence to MADP, and 1508 had moderate adherence, as expected in cohorts where the MedDiet is the most common dietary pattern. Only 93 subjects in the entire cohort had a low MADP score, a higher BMI (28.85 ± 3.87) and lower rMED than the other groups. The same group consisted almost entirely of men (89 vs. 4 women) consuming the highest amounts of alcohol, expressed as g alcohol/d in all forms: wine, beer, and spirits. While the women were evenly distributed between moderate and high adherence to MADP, most of the men (57%) had high adherence to MADP. The distribution of participants by scoring criteria of MADP score is shown in Appendix A.

The Charlson Comorbidity Index (CCI) was used to evaluate the impact of comorbidities on mortality prediction [28]. 

A description of each cohort and a comparison among subjects with complete and incomplete data are shown in Appendix A.

The results of the flexible parametric survival analysis models are shown in Table 2, Table 3 and Table 4. We used subjects with high adherence to MADP as the reference category. 

The category with low MADP adherence was associated with a statistically significant higher risk of death from DS (SHR 2.77 95% CI 1.16; 6.63) and cancer mortality (SHR 2.77 95% CI 1.16; 6.63). No statistically significant effect was found for CVD or DOC mortality (Table 2).

A statistically significant positive effect on all causes of death was observed for low adherence to rMEDNA, compared to high adherence (SHR 1.83, 95% CI: 1.00; 3.40). The subgroup with low adherence also had a statistically significant higher risk for cancer mortality (SHR 1.97, 95% CI: 1.06; 3.65). No statistically significant effect was found for mortality from CVD, DS and DOC (Table 3).

The results of mortality hazard ratios (HR) and subdistribution hazard ratios (SHR), according to the combined MADP and rMEDNA categories, are shown in Table 4.

Overall, the deviation from the maximum adherence of the rMEDNA and MADP indices is associated with higher mortality for many of the causes of death examined. Three statistically significant effects were observed for all causes of death: the first for the combination of MADP Moderate and rMEDNA low (HR 1.50, 95% CI: 1.01; 2.23), and the second for MADP Low and rMEDNA Low (HR 2.29, 95% CI: 1.04; 5.04).

Regarding digestive system mortality, there was a statistically significant effect for the combination of MADP Low and rMEDNA Low (SHR 4.38, 95% CI: 1.22; 15.8).

For mortality for cancer other than gastrointestinal, there was a statistically significant effect for the combination of MADP Moderate and rMEDNA Low (SHR 2.82, 95% CI: 1.21; 6.55), and for MADP Low and rMEDNA Moderate (SHR 4.87 95% CI: 1.71; 15.8). No statistically significant effect was found for mortality from other causes or CVD. The modelling process results are graphically represented in Figure 2 and Figure 3.

For all causes of mortality, we observed that the HR curve relating high adherence to the MADP score tends to diverge over time from the SHR curve relating low adherence to the score.

This behavior of the two curves was found also for rMEDNA and the MADP-rMEDNA combination. This divergence is observed starting from death at 60 years of age, and the distance between the two curves increases over time. 

Subdistribution hazard expresses the acceleration at which the event of interest occurs.

For cancer and DS mortality, we observed that the SHR curve relating high adherence to the MADP score tends to diverge over time from the SHR curve relating low adherence to the score.

This behavior of the two curves was also found for rMEDNA and the MADP-rMEDNA combination. This divergence is observed after 50 years, and the distance between the two curves increases over time. For the sake of completeness, we estimated the subdistribution hazard ratio (SHR) for the MADP, rMEDNA and MADP + rMEDNA association with the risk of cancer overall (Appendix A).

## 4. Discussion

Our study investigated the relationship between alcohol consumption patterns, adherence to the MedDiet, and their combined effects on all-cause, cardiovascular, cancer, and digestive-system diseases, and other causes of death. Our findings show that the effects of alcohol consumption vary, depending on drinking patterns and the dietary context in which the alcohol was consumed.

Low MADP adherence was associated with a higher risk of all-cause mortality and cancer mortality, as in previous studies [11]. In our results, low MADP adherence was also associated with a higher risk of digestive system causes of deaths, which included both neoplastic and non-neoplastic causes.

By combining the effects of adherence to MADP and the Mediterranean diet, we found that overall mortality risk, and especially digestive mortality risk, were associated with the categories of low adherence to MADP and low adherence to rMEDNA, showing a synergy between the two indicators. The risk of cancer mortality, excluding gastrointestinal cancers, was increased in the categories of moderate adherence to MAPD and low adherence to the Mediterranean diet, and low adherence to MADP and moderate adherence to the Mediterranean diet. In contrast, previous studies had analyzed the effect of diet and MADP index on mortality, without finding a synergy between the combined effects.

It was not the objective of the study to establish a causal link, but our results confirm a synergy between diet and alcohol, hypothesized in several studies [29,30], and attribute a value to this interaction in influencing the risk of death. In our results, no significant effects were found for mortality from cardiovascular diseases and from other causes.

We chose to use the MADP index rather than ethanol quantities, as it better investigated different patterns of alcohol consumption. There is strong evidence supporting recommendations on alcohol consumption varying by geographic location [10]. The MADP index assessed aspects such as the amount, type and context of alcohol consumption, which can determine the effect of alcohol on health. 

In our cohort, there was a widespread pattern of moderate consumption of wine during meals, typical of southern Italian populations and the MedDiet. The type of alcoholic beverage could influence the effect of alcohol on health, although not all authors are in agreement on this. Studies conducted on different types of alcoholic beverages have shown that moderate wine consumption is associated with a lower risk of cardiovascular disease [31,32], cancer [31,33,34] and all-cause mortality [31,35,36], as compared to beer and spirit consumption. 

Our results were consistent with these expectations, showing lower mortality among wine drinkers compared to drinkers of other alcohols. This protective benefit may be due in part to the lower total amount of alcohol consumed, but a protective benefit of wine could be reinforced by the presence of significant amounts of polyphenols and bioactive components in wine, such as catechin, quercetin, anthocyanins and resveratrol [37].

This is one of the most important differences between wine and other alcoholic beverages. In fact, red wine contains on average 1.8 g/L of polyphenols, and the content in white wine ranges between 0.2 and 0.3 g/L [38,39].

These beneficial substances could interact with those of the MedDiet to mitigate the negative effects of alcohol. Indeed, the MedDiet is characterized by high intakes of foods rich in polyphenols, such as fresh fruit, vegetables, legumes, whole grains, nuts, and extra-virgin olive oil, and it shows improved markers of inflammation and oxidative stress and improved insulin sensitivity, lipid profile and endothelial function and antithrombotic properties. Most of these effects could be attributable to bioactive ingredients, including fiber, polyphenols, and mono- and polyunsaturated fatty acids [40].

It is also known that wine drinkers have a lower risk of becoming heavy drinkers [41] and are less exposed to binge drinking, which in other studies was widespread, especially among young people, and which has been shown to be an important risk factor for mortality [42]. Heavy episodic drinking at least once a month was associated with fibrosis progression in patients with metabolic-associated fatty liver disease [43].

The MedDiet has been extensively studied as a means to reduce the incidence or mortality from cancer [44]. In our results, this effect was observed in the subgroup of cancers but not in the digestive system causes of deaths. This may be due to the fact that in the group of causes of death related to the DS, causes directly or indirectly related to hepatitis virus infections (viral hepatitis, virus-related cirrhosis, and hepatocellular carcinoma) are highly represented, on which the Mediterranean diet alone has a poor impact, while the pattern of alcohol intake can have a protective effect against these risks, as alcohol and viruses synergistically cause liver damage.

On the other hand, there is a strong consensus in the literature that alcohol drinking can cause a dose-related increase in cancer risk [5]. Our results suggest that the effect of the diet was combined with the effect of alcohol; in fact, drinkers with low adherence to the MADP and low adherence to the rMEDNA had the highest risk of mortality from all causes and mortality from cancer. Subjects with high adherence to the diet and low adherence to the MADP were too few to estimate the risk of death; this is because, in our cohort, those with good eating habits rarely consumed high quantities of alcohol.

Tobacco smoking was a potential confounder in the study of the relationship between alcohol-consumption patterns and health outcomes, given the strong association with neoplastic and nonneoplastic effects in terms of mortality. [45] The percentage of smokers was higher in the category of low adherence to MADP compared to the others (38% of the total sample vs. 19% for moderate adherence and 14% for high adherence). However, the inclusion of tobacco smoking in the model did not influence the results for the outcomes of greatest interest.

This study has several strengths. Using the MADP index allowed us to incorporate patterns of alcohol consumption and therefore distinguish between individuals who occasionally drank heavily and episodically and those who consumed the same amount of alcohol over several days. Assessing an overall pattern of drinking was found to be better than simply assessing the total amount of alcohol consumption for capturing the potential effect of alcohol consumption on mortality. Furthermore, this is the first time the MADP index has been used in an Italian cohort.

Few previous studies analyzing the effects of alcohol consumption on mortality considered how the relationship between alcohol consumption and health could depend on background rates of the disease.

Furthermore, compared to previous studies [11], we used a competitive risk analysis for the following causes: cardiovascular disease, digestive disorders, cancer, and other causes. The competitive risk methodology allows us to better analyze the impact of different causes on the achievement of a composite end-point when the risk factor is related to several outcomes of interest [46].

This study also has limitations that should be taken into consideration when interpreting the results. We did not have any information on the physical activity of the participants, and the only available proxy for physical activity was the occupational category. Categories such as “craft, agriculture and sales” and “elementary occupations” include manual tasks that lead to higher energy consumption, compared to the others. Furthermore, it was not possible to include the burden of alcohol-abuse disorders. Finally, we preferred to exclude the abstainers group because we could not distinguish between lifelong abstainers (by personal choice) and those who, suffering from an alcohol-related disorder, had been forced to quit alcohol consumption. In the latter subjects, the excess mortality could be due to the alterations that had developed previously, which had led to the elimination of alcohol, and not to the abstinence itself. Many people who stop drinking do so due to health problems, which may or may not be caused by alcohol, and this implies that, according to some studies, people who drink moderate amounts of alcohol are often healthier than non-drinkers (the so-called “quitter effect”) [47].

## 5. Conclusions

In conclusion, low adherence to MADP and low adherence to rMEDNA were associated with higher mortality risk, especially digestive mortality risk. So, high adherence to the MedDiet could mitigate the harmful effects of minimal amounts of alcohol. Our findings highlight the importance of considering alcohol-consumption patterns, type of alcoholic beverages, and diet, together, when evaluating effects on mortality. Understanding the relationship between alcohol and mortality is essential for developing effective public health policies and guidelines to improve public awareness of the risks associated with alcohol consumption. However, further research is needed to better understand the complex relationships between alcohol and diet and mortality.

## Figures and Tables

**Figure 1 nutrients-16-00059-f001:**
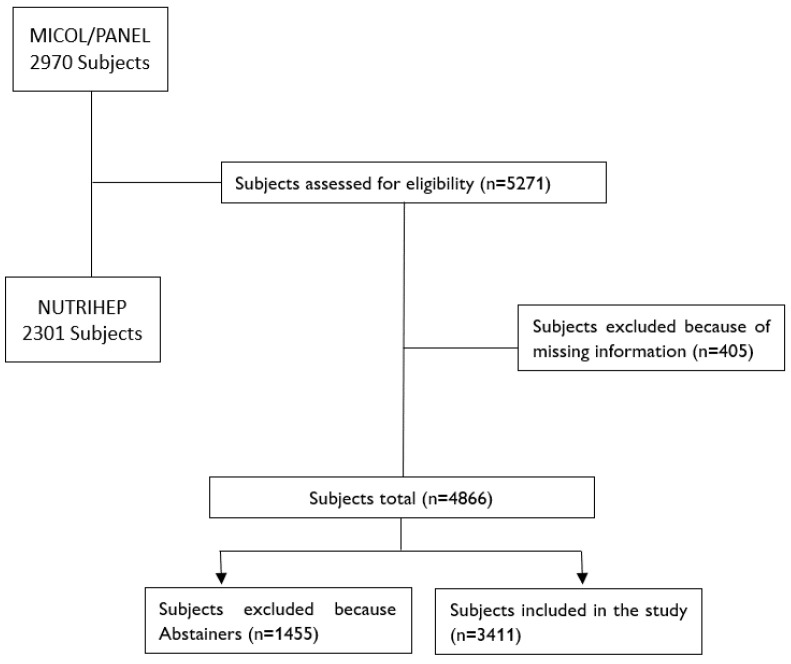
Flowchart of MICOL/PANEL and NUTRIHEP studies and final cohort.

**Figure 2 nutrients-16-00059-f002:**
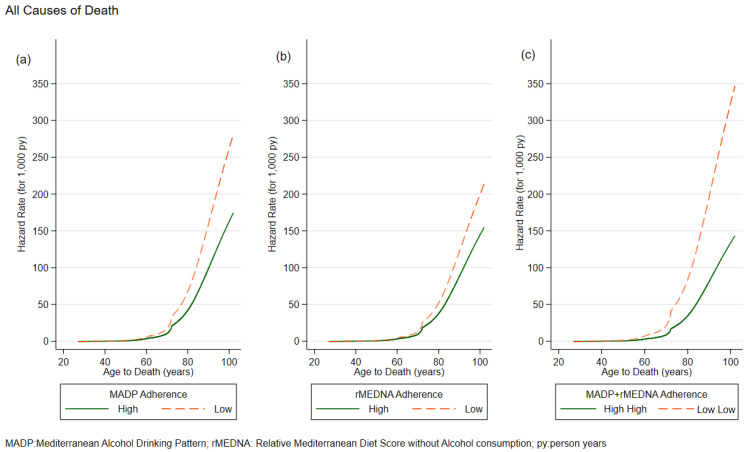
Hazards rates for all causes of death by (**a**) MADP, (**b**) rMEDNA, (**c**) MADP + rMEDNA. MADP: Mediterranean Alcohol Drinking Pattern; rRMDNA: Relative Mediterranean Diet Score without alcohol consumption.

**Figure 3 nutrients-16-00059-f003:**
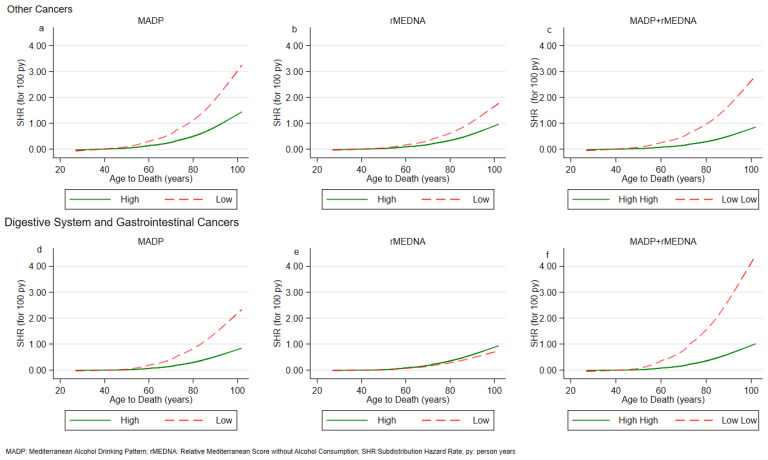
Subdistribution Hazard Rate for Cancer Death by (**a**) MADP, (**b**) rMEDNA, (**c**) MADP + rMEDNA and Digestive system and Gastrointestinal Cancer by (**d**) MADP, (**e**) rMEDNA, (**f**) MADP + rMEDNA. SHR: Subdistribution Hazard Rate; MADP: Mediterranean Alcohol Drinking Pattern; rRMDNA: Relative Mediterranean Diet Score without alcohol consumption.

**Table 1 nutrients-16-00059-t001:** Characteristics of Participants by Mediterranean Alcohol Drinking Pattern (MADP) Categories MICOL/PANEL and NUTRIHEP Studies. Castellana Grotte, Putignano (BA), Italy, 2005–2022.

		Mediterranean Alcohol Drinking Pattern			
	All Sample ^¥^	High (7–9)	Moderate (4–6)	Low (0–3)	*p*-Value
N ***	3411	1810	1508	93	
Age at enrolment (yrs) *	50.98 (16.01)	54.79 (15.62)	46.27 (15.38)	53.16 (13.31)	<0.001
DBP (mmHg) *	124.98 (16.98)	126.09 (17.90)	123.33 (15.53)	129.92 (18.32)	<0.001
SBP (mmHg) *	77.59 (9.39)	77.08 (9.91)	78.09 (8.67)	79.41 (9.29)	0.002
Weight (kg) *	72.98 (14.52)	73.50 (14.22)	71.86 (14.78)	81.26 (12.78)	<0.001
BMI (kg/m^2^) *	27.13 (4.80)	27.57 (4.67)	26.50 (4.92)	28.85 (3.87)	<0.001
Kcal days *	2185.4 (722.8)	2165.5 (682.8)	2168.4 (746.7)	2849.4 (778.6)	<0.001
Triglycerides (mmol/L) *	1.38 (0.98)	1.42 (0.95)	1.31 (0.99)	1.61 (1.13)	<0.001
Total Cholesterol (mmol/L) *	5.12 (1.02)	5.18 (0.99)	5.03 (1.06)	5.43 (0.98)	<0.001
HDL (mmol/L) *	1.34 (0.35)	1.33 (0.34)	1.34 (0.34)	1.41 (0.48)	0.099
LDL (mmol/L) *	3.16 (0.87)	3.21 (0.85)	3.09 (0.89)	3.27 (0.87)	<0.001
Glucose (mmol/L) *	5.84 (1.34)	5.93 (1.33)	5.70 (1.34)	6.25 (1.30)	<0.001
GPT (μkat/L) *	0.27 (0.21)	0.27 (0.23)	0.27 (0.18)	0.37 (0.22)	<0.001
Wine (g alcohol/d) *	15.77 (17.86)	16.47 (11.13)	12.73 (21.39)	51.40 (21.19)	<0.001
Beer (g alcohol/d) *	1.87 (4.22)	0.75 (2.00)	2.42 (4.43)	14.89 (7.69)	<0.001
Spirit (g alcohol/d) *	1.19 (2.97)	0.74 (1.61)	1.39 (3.09)	6.82 (9.03)	<0.001
Gender ***					
Female	1464 (42.9)	701 (47.9)	759 (51.8)	4 (0.3)	<0.001
Male	1947 (57.1)	1109 (57.0)	749 (38.5)	89 (4.6)	
Smoker ***					
Never/Former	2823 (82.8)	1550 (54.9)	1216 (43.1)	57 (2.0)	<0.001
Current	588 (17.2)	260 (44.2)	292 (49.7)	36 (6.1)	
CCI **	2 (1–4)	3 (1–5)	2 (0–3)	4 (2–5)	<0.001
rMED **	8 (6–10)	8 (7–10)	8 (6–11)	7 (5–8)	<0.001
rMEDNA categories					
High Adherence	654 (19.2)	354 (54.1)	293 (44.8)	7 (1.1)	<0.001
Moderate Adherence	1697 (49.8)	996 (58.7)	658 (38.8)	43 (2.5)	
Low Adherence	1060 (31.1)	460 (43.4)	557 (52.5)	43 (4.1)	
Age at Death (yrs) **	67.4 (55.0–78.3)	71.6 (58.7–81.6)	61.8 (50.9–74.0)	69.1 (59.0–77.07)	<0.001
Observation time ** (yrs)	16.8 (16.1–17.0)	16.8 (16.0–17.1)	16.8 (16.2–16.9)	16.8 (16.0–17.2)	0.003
Status ***					
Alive and/or Censored	2819 (82.6)	1406 (49.9)	1345 (47.7)	68 (2.4)	<0.001
Dead	592 (17.4)	404 (68.2)	163 (27.5)	25 (4.2)	
Cause of Death ***					
CVD	175 (29.6)	126 (72.0)	44 (25.1)	5 (2.9)	0.026
Cr	111 (18.8)	71 (64.0)	31 (27.9)	9 (8.1)	
DS	83 (14.0)	49 (59.0)	27 (32.5)	7 (8.4)	
DOC	223 (37.7)	158 (70.9)	61 (27.4)	4 (1.8)	
Education ***					
Primary School	1021 (29.9)	591 (57.9)	403 (39.5)	27 (2.6)	<0.001
Secondary School	1061 (31.1)	563 (53.1)	460 (43.4)	38 (3.6)	
High School	1005 (29.5)	463 (46.1)	522 (51.9)	20 (2.0)	
Graduate	324 (9.5)	193 (59.6)	123 (38.0)	8 (2.5)	
Job ***					
Managers and Professionals	216 (6.3)	96 (44.4)	112 (51.9)	8 (3.7)	<0.001
Craft, Agricultural and Sales Workers	814 (23.9)	401 (49.3)	384 (47.2)	29 (3.6)	
Elementary Occupations	799 (23.4)	429 (53.7)	351 (43.9)	19 (2.4)	
Housewife	415 (12.2)	200 (48.2)	207 (49.9)	8 (1.9)	
Pensioner	1164 (34.2)	619 (65.9)	295 (31.4)	26 (2.8)	
Unemployed	216 (6.3)	64 (28.6)	157 (70.1)	3 (1.3)	
Marital Status ***					
Single	561 (16.4)	212 (37.8)	344 (61.3)	5 (0.9)	<0.001
Married/Cohabiting	2611 (76.5)	1451 (55.6)	1082 (41.4)	78 (3.0)	
Separated/Divorced	73 (2.1)	28 (38.4)	40 (54.8)	5 (6.8)	
Widower	166 (4.9)	119 (71.7)	42 (25.3)	5 (3.0)	

DBP: Diastolic Blood Pressure; SBP: Systolic Blood Pressure; BMI: Body Mass Index; HDL: High Density Lipoprotein Cholesterol; LDL: Low Density Lipoprotein Cholesterol; GPT: Glutamate Pyruvate Transaminase; CCI: Comorbidity Charlson Index; rMED: Relative Mediterranean Score; rMEDNA: Relative Mediterranean Score without alcohol consumption; CVD: Cardiovascular Disease; Cr: Cancer deaths; DS: Digestive System and gastrointestinal cancer deaths; DOC: Deaths from Other Causes. Cells reporting subject characteristics contain * Mean ± (SD), ** Median (IQR), *** Number, (Percentage) Percentages calculated for the row. ^¥^ Percentages calculated for the column.

**Table 2 nutrients-16-00059-t002:** Mortality Hazard Ratios (HR) and Subdistribution Hazard Ratios (SHR), according to the categories of the Mediterranean Alcohol-drinking pattern (MADP).

	MADP	
	Moderate (4–6)	Low (0–3)
	HR (95% CI)	HR (95% CI)
All causes of Death	1.09 (0.89; 1.34)	1.46 (0.93; 2.28)
	SHR (95% CI)	SHR (95% CI)
CVD	1.09 (0.74; 1.59)	0.81 (0.30; 2.19)
Cr	094 (0.60; 1.45)	2.25 * (1.08; 4.70)
DS	1.58 (0.94; 2.65)	2.77 * (1.16; 6.63)
DOC	1.08 (0.76; 1.53)	0.52 (0.17; 1.62)

High Adherence: reference category; * *p* value < 0.05. Model adjusted for gender (F vs. M), BMI: Body Mass Index (<25 vs. ≥25), GPT: Glutamate Pyruvate Transaminase (<40 µ/L vs. ≥40 µ/L), Triglycerides (<150 mg/L vs. ≥150 mg/L), Comorbidity Charlson Index, Smoking habits, Job. HR: Hazard Ratio; SHR: Subdistribution Hazard Ratio; CVD Cardiovascular Disease; Cr: Cancer deaths; DS: Digestive System and gastrointestinal cancer deaths; DOC: Deaths from Other Causes.

**Table 3 nutrients-16-00059-t003:** Mortality Hazard Ratios (HR) and Subdistribution Hazard Ratios (SHR), according to the categories of the rMEDNA.

	rMEDNA	
	Moderate (4–6)	Low (0–3)
	HR (95% CI)	HR (95% CI)
All causes	1.13 (0.90; 1.43)	1.27 (0.96; 1.68)
	SHR (95% CI)	SHR (95% CI)
CVD	0.97 (0.65; 1.45)	0.82 (0.49; 1.38)
Cr	1.58 (0.91; 2.75)	1.83 * (1.00; 3.40)
DS	1.07 (0.60; 1.90)	0.79 (0.37; 1.69)
DOC	1.18 (0.80; 1.75)	1.35 (0.86; 2.13)

High Adherence: reference category; * *p* value < 0.05. Model adjusted for gender (F vs. M), BMI: Body Mass Index (<25 vs. ≥25), GPT: Glutamate Pyruvate Transaminase (<40 µ/L vs. ≥40 µ/L), Triglycerides (<150 mg/L vs. ≥150 mg/L), Comorbidity Charlson Index, Smoking habits, Job. HR: Hazard Ratio; SHR: Subdistribution Hazard Ratio; CVD Cardiovascular Disease; Cr: Cancer deaths; DS: Digestive System and gastrointestinal cancer deaths; DOC: Deaths from Other Causes.

**Table 4 nutrients-16-00059-t004:** Mortality Hazard Ratios (HR) and Subdistribution Hazard Ratios (SHR), according to the categories of the MADP + rMED No Alcohol.

	All Causes	CVD	DS	Cr	DOC
	HR (95% CI)	SHR (95% CI)	SHR (95% CI)	SHR (95% CI)	SHR (95% CI)
MADP#rMEDNA					
High–High	1.00	1.00	1.00	1.00	1.00
High–Moderate	1.24(0.93; 1.65)	1.13 (0.69; 1.85)	0.85 (0.42; 1.74)	2.04 (0.99; 4.18)	1.31 (0.82; 2.12)
High–Low	1.26 (0.87; 1.22)	0.83 (0.42; 1.65)	0.62 (0.21; 1.78)	1.67 (0.71; 3.96)	1.59 (0.90; 2.79)
Moderate–High	1.37 (0.90; 2.08)	1.45 (0.70; 3.00)	1.18 (0.40; 3.46)	1.52 (0.56; 4.11)	1.46 (0.71; 3.02)
Moderate–Moderate	1.17 (0.82; 1.66)	1.01 (0.54; 1.88)	1.70 (0.77; 3.74)	1.04 (0.42; 2.57)	1.29 (0.72; 2.31)
Moderate–Low	1.50 * (1.01; 2.26)	1.13 (0.54; 2.36)	0.77 (0.24; 2.45)	2.82 * (1.21; 6.55)	1.48 (0.75; 2.89)
Low–High	0.57 (0.08; 4.12)	1.71 (0.23; 12.8)	NE	NE	NE
Low–Moderate	1.78 (0.97; 3.27)	0.69 (0.16; 2.95)	1.74 (0.47; 6.42)	4.87 * (1.71; 13.9)	0.73 (0.17; 3.09)
Low–Low	2.29 * (1.04; 5.04)	0.79 (0.11; 5.63)	4.38 * (1.22; 15.8)	3.33(0.72; 15.4)	0.72 (0.10; 5.05)

* *p* < 0.05. Model adjusted for gender (F vs. M), BMI (<25 vs. ≥25), GPT (<40 vs. ≥40), Triglycerides (<150 vs. ≥150), Comorbidity Charlson Index, Job, Smoking habits. HR: Hazard Ratio; SHR: Subdistribution Hazard Ratio; CVD Cardiovascular Disease; Cr: Cancer Deaths; DS: Digestive System and Gastrointestinal Cancer Deaths; DOC: Deaths from Other Causes; NE: Not Estimable.

## Data Availability

The original contributions submitted in the study are incorporated in the article. Additional enquiries may be addressed to the corresponding author.

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
