# Peer review of "High Adherence to a Mediterranean Alcohol-Drinking Pattern and Mediterranean Diet Can Mitigate the Harmful Effect of Alcohol on Mortality Risk"

_nutrients, 2023, doi:10.3390/nu16010059_

Round 1

Reviewer 1 Report

Comments and Suggestions for Authors

The study presented by Campanella and colleages (2023) aimed to investigate the joint effect of adherence to the MedDiet and alcohol-drinking pattern on all-cause mortality, incluiding cardiovascular, digestive system and cancer. For this purpose, they used data from 3411 participants from two different prospective cohort study (MICOL/PANEL and NUTRIHEP).

The introduction section  discusses the role of lifestyle and socio-demographic factors on the effects of moderate alcohol consumption. However, these are not addressed in the study itself. For example, no data are given on physical exercise, and despite including measures of tobacco consumption, they are not discussed. It should not be overlooked that people who drink more moderately (or even not at all) are likely to be the same people who, in general, will lead a healthier lifestyle. Are the results due to the pattern of alcohol consumption, to the lifestyle or to the synergy of both?

It is necessary to include a paragraph explaining what this study contributes to all the similar studies that already exist. Otherwise, it is difficult to understand the relevance of this study. Explaining specifically what the study contributes to the state of the art on this issue is essential.

Line 62-63: Please reconsider rephrasing the last part of the objective as follows “…MADP index and all-cause mortality, which includes cardiovascular disease, digestive disorders, cancer and other causes.”

The paragraph beginning on line 103 should be revised. Some words and punctuation are missing.

The authors use the cut-off defined by Gea et al. (2014) to assign MADP scores. However, However, the way of grouping them into categories is different. Please justify this and point out on which articles they rely on to do it this way.

Reviewing the characteristics of participants by MADP categories in NUTRIHEP study and MICOL/PANEL study there are some disparate data (table S1 and S2), to what could this be due? In addition, the disaggregated data for DS would be missing. Have you considered to what extent these differences in the distribution of the different categories might distort the overall data?

On the other hand, you write 9 lines (from 128 to 135) to describe rMED and the way it is scored. Wouldn't it be better to talk directly about rMEDNA, which is the one you are going to use? This would avoid misleading data such as the different categories indicated in line 135 and line 160.

Within the statistical analysis section the authors indicate that they will use some variables as potential confounders without justifying why. Just to give an example, why is job or occupation a cofounding factor and not educational level? Just to give an example, why is job or occupation a cofounding factor and not educational level? In relation to this, please review tables S5 and S6 as it appears that the model has been fitted for education and not for job (which is what is pointed out in the text and in table 2 and 3).

Please, not to confuse alcohol consumption with drinking patterns (line 282). In the study, the amount of alcohol consumed has not been used as a variable, but the type of alcohol consumption followed by the participants in relation to their MADP scores. On another note, I fully understand the relevance of including the consumption pattern as an important factor. However, I believe it is essential in this type of study to include abstaining participants.  Despite the limitations that the authors themselves indicate, they would be a control group to be considered. If we continue with the same justification, the present study should have asked the participants how long they have been following the Mediterranean pattern of alcohol consumption.

Minor changes:

·         Include legend of figure 1. And delete the word “flowchart” in line 83

      There are some abbreviations that are not defined in their first appearance in the text (line 78: GPs; line 99: BP; line 182: GPT and TGL)

·         Revise table 3 and 6S, as the cut-off points of the categories do not correspond to what appears in the written text. For example, participant classified as moderate adherence to rMEDNA scored 7-10 and not 4-6.

·         Please, provide a bibliographic citation for the following sentence “…olive oil, of which Italy is a major producer and consumer” (line 139-140)

Reviewer 2 Report

Comments and Suggestions for Authors

I propose to replace the word alcohol with ethanol.

There is inconsistency between the objectives and the conclusions.

Author Response

-I propose to replace the word alcohol with ethanol.

We have included it in the introduction:

 Alcohol, like ethanol, is a psychoactive substance with a significant impact on public health and mortality.

-There is inconsistency between the objectives and the conclusions.

we have taken your recommendation and we have modified the manuscript

Reviewer 3 Report

Comments and Suggestions for Authors

This interesting study used dietary and alcohol intake data from two prospective cohorts of Italian adults to examine associations between adherence to the Mediterranean diet (MedDiet) and Mediterranean Alcohol-drinking Pattern Index (MADP) and all-cause mortality, and mortality from cardiovascular disease, cancer, digestive system diseases, and other causes. Analysis results indicated greater risk for all-cause mortality, cancer, and digestive system mortality among individuals with low adherence to the MADP. Greater risk of overall mortality and digestive system diseases was also observed among individuals with low adherence to the MADP and MedDiet. These findings suggest that potentially negative effects of alcohol consumption on chronic illnesses can be mitigated by adherence to a Mediterranean diet and moderate levels of wine consumption, particularly red wine that is higher in polyphenols.

Regarding study methods, was the EPIC administered at each round of data collection? If so, how were the MADP and rMED scores determined with multiple EPIC assessments? To what extent does the EPIC represent an accurate assessment of dietary and alcohol intake patterns over many years? If the EPIC does not accurately represent dietary and alcohol intake patterns over many years then this should be mentioned as a study limitation.

The authors note that physical activity was not assessed, and this should also be mentioned as a study limitation, though factors such as BMI and cholesterol levels could be considered as proxies for physical activity.

Footnotes for Table 2 and Table 3 should mention potential confounding variables included in the models (e.g., gender, BMI). 
